# Quantitative Structure–Activity Relationships for the Flavonoid-Mediated Inhibition of P-Glycoprotein in KB/MDR1 Cells

**DOI:** 10.3390/molecules24091661

**Published:** 2019-04-27

**Authors:** Mengmeng Xia, Yajing Fang, Weiwei Cao, Fuqiang Liang, Siyi Pan, Xiaoyun Xu

**Affiliations:** Key Laboratory of Environment Correlative Dietology (Huazhong Agricultural University), Ministry of Education, Wuhan 430070, China; xiamengmeng@webmail.hzau.edu.cn (M.X.); fyj@webmail.hzau.edu.cn (Y.F.); weiweicao@webmail.hzau.edu.cn (W.C.); lfqzyw@webmail.hzau.edu.cn (F.L.); Pansiyi@mail.hzau.edu.cn (S.P.)

**Keywords:** flavonoids, P-gp, inhibitory activity, QSAR

## Abstract

P-glycoprotein (P-gp) serves as a therapeutic target for the development of inhibitors to overcome multidrug resistance (MDR) in cancer cells. In order to enhance the uptake of chemotherapy drugs, larger amounts of P-gp inhibitors are required. Besides several chemically synthesized P-gp inhibitors, flavonoids as P-gp inhibitors are being investigated, with their advantages including abundance in our daily diet and a low toxicity. The cytotoxicity of daunorubicin (as a substrate of P-gp) to KB/MDR1 cells and the parental KB cells was measured in the presence or absence of flavonoids. A two-dimensional quantitative structure–activity relationship (2D-QSAR) model was built with a high cross-validation coefficient (Q^2^) value of 0.829. Descriptors including vsurf_DW23, E_sol, Dipole and vsurf_G were determined to be related to the inhibitory activity of flavonoids. The lack of 2,3-double bond, 3′-OH, 4′-OH and the increased number of methoxylated substitutions were shown to be beneficial for the inhibition of P-gp. These results are important for the screening of flavonoids for inhibitory activity on P-gp.

## 1. Introduction

P-glycoprotein (P-gp) is a member of the ATP-binding cassette (ABC) family, which is encoded by the *ABCB1* gene. The overexpression of P-gp has been associated with multidrug resistance (MDR) in cancer cells, a limiting factor in the success of cancer chemotherapy. Given the role of P-gp in influencing cancer chemotherapy, methods to overcome P-gp-mediated efflux have been investigated. It is generally believed that the mechanisms of P-gp inhibition mainly comprise four aspects: competitively, non-competitively or allosterically blocking the drug binding site; interfering with the ATP hydrolysis process; altering the integrity of cell membrane lipids; decreasing the P-gp expression [1]. Four generations of P-gp inhibitors have been identified in recent years. The first-generation inhibitors, including verapamil [2] and cyclosporine A [3], were found to possess high toxicity at their effective doses [4]. The derivatives of the first-generation inhibitors, dexverapamil and VX710, are termed the second-generation inhibitors. However, due to their impact on P450 and drug interaction profiles, these inhibitors were not used clinically [5]. Elacridar, tetrandrine, and zosuquidar, the third-generation inhibitors, are limited due to their low survival [6]. Therefore, high-potency and low-toxicity P-gp inhibitors are urgently required for chemotherapy treatment. Compounds from natural products belonging to the fourth-generation P-gp inhibitors are of great significance [7].

Flavonoids are a class of compounds based on the diphenylpropane (C6–C3–C6) skeleton, which are widespread in our common diet, including in fruits and vegetables. Flavonoids have been shown to be beneficial to human health for their antioxidant, anti-inflammatory, anticarcinogenic and antiviral activities [8]. Several studies have suggested that flavonoids can inhibit P-gp in order to enhance the bioavailability and uptake of anticancer drugs [9]. Flavonoids (biochanin A, morin [10], silymarin [10,11], quercetin [11,12,13], kaempferol [11,12,13,14] and tangeretin [15]) have been demonstrated to present inhibitory activity on P-gp. Kitagawa [16], by studying the structure–activity relationships (SARs) of flavonoids, found that the planar structure and hydrophobic nature of flavonoids are important for the inhibitory effect on P-gp. Quantitative structure–activity relationships (QSARs) can be used for observing the mechanisms between molecular structures and various biological activities [17]. QSAR has been widely used to determine whether a compound is an inhibitor of P-gp. Various studies have built the three-dimensional quantitative structure–activity relationships (3D-QSAR) model to investigate the inhibitory activity on P-gp [18,19,20]. The model is limited as it is based on the assumption that compounds all act on the same receptor. Furthermore, the 3D-QSAR model is affected by the quality of molecular alignments/superimpositions and information on ligand bioactive conformations [21]. The two-dimensional quantitative structure–activity relationships (2D-QSAR) model does not require subjective (or time-consuming) molecular alignment or putative binding conformation or determination of 3D structures. Furthermore, 2D-QSAR is simple and robust but has been rarely reported.

The aim of this study was to investigate the quantitative structure–activity relationship for the flavonoid-mediated inhibition of P-gp in KB/MDR1 cells overexpressed with P-gp. Daunorubicin [22] has been reported to be an anticancer drug and the substrate P-gp. In this study, the inhibitory activity (IC_50_ of daunorubicin) of 31 flavonoids (Table 1) was measured and used to build the 2D-QSAR model to determine the relationship between flavonoid structure and inhibitory activity. The structure characteristics which interact with P-gp could enhance the uptake of chemotherapy drugs.

## 2. Results

### 2.1. Cytotoxicity

The reversal effect (represented by IC_50_) of flavonoids on the cytotoxicity of daunorubicin to KB and KB/MDR1 cells is shown in Table 2. A decrease in IC_50_ value indicates a higher flavonoid inhibitory activity. Elacridar (10 μM) was used as a positive control. The IC_50_ values of daunorubicin as a negative control (without any inhibitors) in KB cells were significantly lower compared to those in KB/MDR1 cells, suggesting that daunorubicin is less sensitive to KB/MDR1 cells due to the overexpression of P-gp. Elacridar significantly enhanced the cytotoxicity of daunorubicin in KB/MDR1 cells, being more pronounced in KB/MDR1 cells (RF_KB/MDR1_ = 6.818) compared to KB cells (RF_KB_ = 1.512). As shown in Table 2, the KB/MDR1 cells showed 10-fold and 21-fold resistance to daunorubicin compared to the KB cells of **17** (quercetin) and **28** (daidzein), respectively, indicating that quercetin and daidzein can increase the sensitivity of KB/MDR1 cells to anticancer drug substrates. However, the fact that the RF_KB/MDR1_ of quercetin and daidzein was 0.580 and 0.704, respectively, indicated an increased P-gp function, while their RFKB was as high as 10.414 and 3.531. These results may be due to the fact that although there is no P-gp expression in KB cells, expression of other proteins like multidrug resistance protein (MRP), or breast cancer resistance protein (BCRP) may occur, which also have the same function as P-gp.

In KB/MDR1 cells, flavonoid **9** showed the highest RF_KB/MDR1_, 4.586, followed by **8** and **4** with RF_KB/MDR1_ values of 3.613 and 3.443, respectively. As we know, flavonoids with an RF_KB/MDR1_ greater than 1.000 are potential inhibitors of P-gp. However, as shown in Table 2, the RF_KB/MDR1_ of flavonoids **11**, **12** and **29** is less than 1.000; these results suggested that luteolin, vitexin and puerarin are potential activators of P-gp.

Flavonoids **23**, **25** and **26** which lack the 2,3-double bond in the C ring, had a lower IC_50_ than their corresponding flavone or flavonol. These findings indicated that the 2,3-double bond is unfavorable to the inhibition of P-gp. Comparing the IC_50_ of flavonoids **1** (2.373 μM, R_5_=OMe), **2** (1.579 μM, R_5_, R_7_ =OMe), **3** (1.580 μM, R_5_, R_3′_=OMe), and **4** (0.901 μM, R_5_, R_7_, R_3′_=OMe), we found that with an increased number of methoxylated substitutions, an increased inhibitory activity on P-gp occurred. The structure–affinity relationship implicated that 3′-OH and 4′-OH are not conducive to the inhibitory activity of flavonoids by comparing **7** (R_3′_, R_4′_ =H, inhibitor) and **11** (R_3′_, R_4′_=OH, non-inhibitor). It was also supported by an inhibitor galangin (R_3′_, R_4′_=H) and a non-inhibitor quercetin (R_3′_, R_4′_=OH).

### 2.2. QSAR Study

The model was obtained utilizing IC_50_ as the dependent variable and the calculated molecule descriptors as independent variables. The best QSAR model established using a training set consisting of 24 flavonoids and a test set of seven flavonoids is as follows:IC_50_ = 0.183vsurf_DW23 − 0.359E_sol − 3.181dipole + 10.627vsurf_G − 9 .859R^2^ = 0.892, R_adj_^2^ = 0.869, Q^2^ = 0.829, F = 39.073, *p* < 0.01, RMSE = 0.492, R_pred_^2^ = 0.905

The correlation matrix between IC_50_ and related molecular descriptors is shown in Table 3. The descriptors of vsurf_DW23 and E_sol are significantly related to IC_50,_ indicating that vsurf_DW23 and E_sol play an important role in the inhibitory activity of flavonoids. Also, the Pearson correlation coefficient |r| < 0.5 between each descriptor indicates that the model has not been over-fitted. The square of the correlation coefficient between the experimental and predicted IC_50_ values reached 0.904 (Figure 1), indicating that the experimental value is consistent with the predicted value. In addition, the model was verified by cross-validation (leave-one-out), and the cross-validation coefficient (Q^2^) was as high as 0.829, suggesting that the obtained model has great prediction ability. The test set prediction correlation coefficient reached 0.905, indicating that the model has better external prediction ability.

The QSAR model contains two molecular descriptors, E_sol and dipole, which are positively correlated with the inhibitory activity of flavonoids as well as two negatively correlated descriptors, vsurf_DW23 and vsurf_G.

Vsurf_DW23 and vsurf_G are the vsurf_descriptors, which are similar to the VolSurf descriptors. Vsurf_DW23 is a hydrophobic zone parameter that represents the contact distance of vsurf_EWmin (lowest hydrophilic energy) when a water probe interacts with a target molecule [23]. Vsurf_DW23 is positively correlated to IC_50_, indicating that this descriptor related negatively to the inhibitory activity of flavonoids on P-gp. With increasing IC_50_ value, the inhibitory activity of flavonoids decreased. As shown in Table 4, it was observed that flavonoids with a vsurf_DW23 value above 10, i.e., **11** (14.221), **17** (11.597), and **28** (16.523), showed high IC_50_.

The positive contribution of E_sol is the potential energy descriptor that represents the solvation energy. The influence on IC_50_ of E_sol is illustrated by comparing the E_sol and IC_50_ of flavonoid **16** (1.410, 2.463 μM) (Table 4) versus **20** (−0.215, 2.721 μM). Moreover, flavonoid **12**, with a low vsurf_DW23 value (1.118), exhibited a high IC_50_ (5.501 μM) because of the lowest E_sol value (−11.881). We observed that the higher the E_sol value, the smaller the IC_50_.

Dipole is the conformation-dependent charge descriptor that represents the dipole moment calculated from the partial charges of the molecule. It displays the distribution of charge and the separation degree of negative charge and positive charge [24]. Flavonoid **27** showed a low IC_50_, but exhibited the highest dipole value due to its structure with four chiral centers. Dipole was positively correlated to IC_50_, but negatively correlated to the inhibitory activity, as shown in the model.

Vsurf_G is a shape descriptor describing the surface globularity of the molecule [23]. Flavonoid glycosides including **12**, **13**, **21**, and **29** have a high vsurf_G value. Flavonoid **27** also exhibits a high vsurf_G value related to its special structure. Vsurf_G plays an important role in the inhibitory activity on P-gp.

## 3. Discussion

It is widely reported that the main cause of the low bioavailability of chemotherapy drugs is related to transport of P-gp [25]. The most typical method to characterize the P-gp function is to construct a cell line with the overexpression of P-gp and then to detect inhibitor-mediated differences in the uptake of P-gp substrates. In this study, the positive inhibitor elecridar significantly reduced the sensitivity of KB/MDR1 cells to daunorubicin, even to the same level as KB cells. The results indicated that P-gp was overexpressed in KB/MDR1 cells and functioned normally. This finding is consistent with our previous research [26]. Thus, the evaluation results of the cell are effective.

In the current study, we investigated the cytotoxicity of daunorubicin in KB/MDR1 and KB cells. The KB/MDR1 cells showed a 4.34-fold (Table 2) resistance to daunorubicin compared to the KB cells, when there were no inhibitors, indicating that the uptake of the chemotherapy drug daunourbicin is affected by P-gp. Flavonoids directly or indirectly inhibit P-gp to enhance the bioavailability of daunorubicin. In agreement with previous studies [10,27], flavonoids **13**, **18** and **30** showed high inhibitory activity. Our results also indicated that the KB/MDR1 cells showed a 10-fold resistance to daunorubicin compared to the KB cells exposed to flavonoid **17**, indicating quercetin. The sensitivity of KB/MDR1 cells to anticancer drug substrates was increased, which is consistent with previous studies reporting that quercetin is a potential modulator of P-gp [11,12,13]. Another study [28] showed that methoxylated substitution and its numbers or sites of the rings are important in the reversal of P-gp-mediated MDR, which is also consistent with our observation that an increased number of methoxylated substitutions resulted in an increased inhibitory activity on P-gp.

In order to determine the molecular properties associated with the inhibitory activity, the QSAR model was built. It was observed that the descriptors of vsurf_DW23 (*p* < 0.01) and E_sol (*p* < 0.05) are significantly related to IC_50_ and the Pearson correlation coefficient |r| < 0.5. These results indicated that the model was reliable. The 2D-QSAR showed a high prediction ability and internal stability with Q^2^ value (0.829) > 0.5 [29], R_pred_^2^ value (0.905) > 0.6 [30].

As seen in the model, vsurf_DW23 was confirmed to be an important factor affecting the inhibitory activity of flavonoids. Vsurf_DW23 is a hydrophobic zone parameter that represents the contact distance of vsurf_EWmin (lowest hydrophilic energy) when a water probe interacts with a target molecule [23]. The model possessing this descriptor shows that the decreased contact distance between the lowest hydrophilic regions is beneficial to the inhibitory activity. It reveals that the hydrophilic regions should be minimal for better activity. We also found that 3′-OH and 4′-OH are not conducive to the inhibitory activity of flavonoids. Thus, the hydrophilicity of flavonoids is not good for its inhibition of P-gp. Moreover, vsurf_G is one of the negatively correlated descriptors which describes the surface globularity of the molecule. The smaller the molecular surface, the better the inhibitory activity.

Based on previous research in the laboratory [31], we found that E_sol is related to the permeability of flavonoids. E_sol represents the solvation energy. It is known that the compound with higher E_sol is harder to solvate. Thus, the compound with higher E_sol entering the binding domain becomes easier. Flavonoids **10** and **17** were reported to be substrates [32] as well as inhibitors of P-gp [33,34]. Competing with substrates for binding to P-gp is one of the mechanisms of inhibition of P-gp. This finding is consistant with our results which suggested that E_sol is associated with the inhibitory activity of P-gp. In addition, dipole is a positively correlated descriptor which represents the dipole moment calculated from the partial charges of the molecule.

In conclusion, flavonoids that had high inhibitory activity on P-gp should have small values of vsurf_DW23 and vsurf_G and large values of E_sol and dipole. Daunorubicin was used as the substrate of P-gp in order to investigate the inhibitory activity of flavonoids. The absence of 2,3-double bond and both 3′-OH and 4′-OH, and an increased number of methoxylated substitutions were shown to be beneficial to P-gp inhibition. Therefore, specific flavonoids meet these criteria to be modulators of P-gp.

## 4. Materials and Methods

### 4.1. Reagents

3-(4,5-dimethylthiazol-2-yl)-2 and 5-diphenylte trazolium bromide (MTT) were purchased from Gen-View Scientific Inc. (Calimesa, CA, USA). Flavonoids were purchased from Aladdin Chemistry Co., Ltd. (Shanghai, China) (purity > 98%). Elacridar was purchased from MedChemExpress. (New Jersey, USA) (purity > 98%) (CAS: 143664-11-3) and used as an inhibitor of P-gp. Daunorubicin hydrochloride was purchased from Tokyo Chemical Industry Co. Ltd. (Tokyo, Japan) (purity > 98%) (CAS:23541-50-6) and used as a substrate of P-gp. All the compounds were dissolved and diluted to a final concentration of 0.1% (*v*/*v*) DMSO. Fetal bovine serum (FBS) was obtained from Zhejiang Tianhang Biotechnology Co.,Ltd. (Zhejiang, China). Dulbecco’s modified essential medium (DMEM) was purchased from Hyclone (Logan, UT, USA). The 96-well plates were purchased from Corning Costar (Cambridge, MA, USA).

### 4.2. Cell Culture

The KB cell line was purchased from Culture Collection of the Chinese Academy of Sciences (Shanghai, China). KB/MDR1 cells were stable, transfected with cDNA of human P-gp, which was confirmed by RT-PCR and Western blotting in our previous study [26]. Both KB and KB/MDR1 cells were grown in DMEM with 10% FBS, penicillin (100 U/mL) and streptomycin (100 μg/mL) in 5% CO_2_, and 90% relative humidity at 37 °C. KB/MDR1 cells were grown with 500 μg/mL hygromycin (Roche, Switzerland).

### 4.3. Cytotoxicity Assay

The cytotoxicity of flavonoids and daunorubicin was determined by MTT assay. The cells were seeded in 96-well plates at a density of 7 × 10^3^ cells/well and incubated in an atmosphere of 5% CO_2_ at 37 °C for 24 h. In order to test the inhibition of flavonoids, cells were treated with daunorubicin (0–20 μM) in the absence or presence of flavonoids at the non-cytotoxic concentration (40 μM) (Appendix A) for 24 h. The medium was then removed, and the cells were incubated in a serum-free medium containing MTT reagent (0.5 mg/mL) for 4 h. DMSO (150 μL/well) was added to dissolve the formazan crystals. A multiskan spectrum microplate reader (Thermo Labsystems, Waltham, MA, USA) was used for measuring absorbance at 490 nm. Since inhibitors can enhance the toxicity of daunorubicin in KB/MDR1 cells [35], the reversal fold (RF_KB/MDR1_, RF_KB_) of flavonoids was estimated by comparing the IC_50_ (concentration of 50% inhibition) of the daunorubicin in the absence of inhibitors to that obtained in the presence of inhibitors. RF was obtained by comparing the IC_50_ of daunorubicin in the KB/MDR1 cells and the parental KB cells.

### 4.4. 2D-QSAR Study

The structures of the 31 flavonoids were drawn by ChemBioDraw ultra 12.0. The initial optimization of these flavonoids was performed by Sybyl X-2.0 (Tripos Inc., St. Louis, MO, USA). The molecular descriptors were calculated by Sybyl X-2.0 and Molecular Operating Environment (MOE) 2009.

The IC_50_ values of the 31 flavonoids in KB/MDR1 cells, in the range 0.676 to 5.894 (Table 2) were used as the dependent variable for QSAR model construction. Seven flavonoids (**2**, **4**, **9**, **20**, **22**, **27** and **30**) were chosen randomly as the test set; the rest of the flavonoids were used as the training set to construct the model. The best molecular descriptors which filtered through the method of the stepwise multiple linear regression (MLR) were adopted as independent variables to build the model. In order to prevent over-fitting between descriptors, correlation analysis between descriptors was performed, and descriptors with an absolute value of correlation coefficients less than 0.5 were selected [29].

The 2D-QSAR model was constructed utilizing the partial least squares (PLS) in Sybyl X-2.0. In the “leave-one-out” method, the Q^2^ of the resulting model is calculated in order to test the predictive ability of the model. The predicted IC_50_ in both the training and test sets was calculated from the QSAR model. For the model to be considered reliable, Q^2^ should be >0.5 [29] and the predictive correlation coefficient (R_pred_
^2^) for the test set should be >0.6 [30]. The best model should have the smallest standard error of estimate (SEE) and the highest F value [18]. Q^2^ value was calculated by the following equation (1):(1)$$Q^{2}=1-\frac{\sum_{i=1}^{n}{\left(y_{i}-\hat{y_{i}}\right)}^{2}}{\sum_{i=1}^{n}{\left(y_{i}-\bar{y}\right)}^{2}}$$

*n* is the number of the sample in the training set, $y_{i}$ and $\hat{y_{i}}$ are the experimental and predicted IC_50_ of the ith sample, and $\bar{y}$ is the average IC_50_ of all samples.

## Figures and Tables

**Figure 1 molecules-24-01661-f001:**
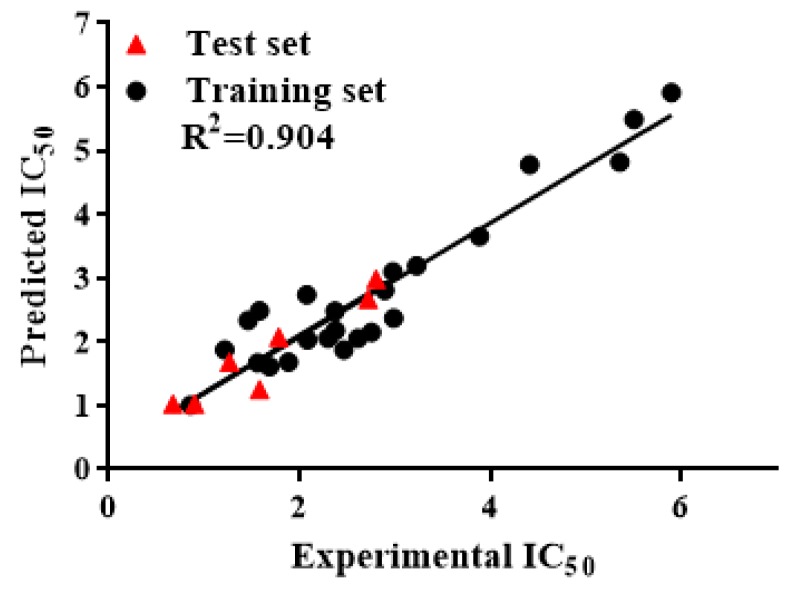
Experimental versus predicted IC_50_ for the model.

**Table 1 molecules-24-01661-t001:** The chemical structures of 31 flavonoids.

No	Flavonoids	CAS	Core Structure	Substructure
**1**	5-Methoxyflavone	42079-78-7	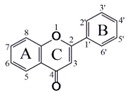	R_5_=OMe
**2**	5,7-Dimethoxyflavone	21392-57-4		R_5_, R_7_=OMe
**3**	5,3′-Dimethoxyflavone			R_5_, R_3′_=OMe
**4**	5,7,3′-Trimethoxyflavone			R_5_, R_7_, R_3′_=OMe
**5**	5,7,3′,4′-Tetramethoxyflavone	855-97-0		R_5_, R_7_, R_3′_, R_4′_=OMe
**6**	Tangertin	481-53-8		R_5_, R_6_, R_7_, R_3′_, R_4′_=OMe
**7**	Chrysin	480-40-0		R_5_, R_7_=OH
**8**	Baicalein	491-67-8		R_5_, R_6_, R_7_ =OH
**9**	Wogonin	632-85-9		R_5_, R_7_ =OH, R_8_=OMe
**10**	Apigenin	520-36-5		R_5_, R_7_, R_4′_=OH
**11**	Luteolin	491-70-3		R_5_, R_7,_, R_3′_, R_4′_=OH
**12**	Vitexin	3681-93-4		R_5_, R_7_, R_4′_ =OH, R_8_=Cglc
**13**	Schaftoside	51938-32-0		R_5_, R_7_, R_4′_ =OH, R_6_=Cglc, R_8_=Carb
**14**	Galangin	548-83-4		R_3_, R_5_, R_7_=OH
**15**	Kaempferide	491-54-3		R_3_, R_5_, R_7_ =OH, R_4′_=OMe
**16**	Fisetin	528-48-3		R_3_, R_7_, R_3′_, R_4′_=OH
**17**	Quercetin	117-39-5		R_3_, R_5_, R_7_, R_3′_, R_4′_=OH
**18**	Morin	480-16-0		R_3_, R_5_, R_7_, R_2′_, R_4′_=OH
**19**	Isorhamnetin	480-19-3		R_3_, R_5_, R_7_, R_4′_ =OH, R_3′_=OMe
**20**	Myricetin	529-44-2		R_3_, R_5_, R_7_, R_3′_, R_4′_, R_5′_=OH
**21**	Rutin	153-18-4		R_3_=ORG, R_5_, R_7_, R_3′_, R_4′_=OH
**22**	Liquiritigenin	578-86-9	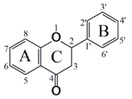	R_7,_, R_4′_=OH
**23**	Naringenin	480-41-1		R_5_, R_7_, R_4′_ =OH
**24**	Hesperetin	520-33-2		R_5_, R_7_, R_3′_ =OH, R_4′_=OMe
**25**	Taxifolin	24198-97-8		R_3_, R_5_, R_7_, R_3′_, R_4′_=OH
**26**	Dihydromyricetin	27200-12-0		R_3_, R_5_, R_7_, R_3′_, R_4′_, R_5′_=OH
**27**	Silibinin	22888-70-6	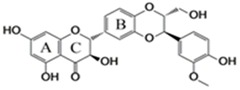	
**28**	Daidzein	40957-83-3	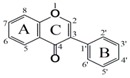	R_7_, R_4′_=OH
**29**	Puerarin	3681-99-0		R_7_, R_4′_ =OH, R_8_=Cglc
**30**	Genistein	446-72-0		R_5_, R_7_, R_4′_=OH
**31**	Biochanin A	491-80-5		R_5_, R_7_ =OH, R_4′_=OMe

Cglc: -C-glucopyranosyl; Carb: -O-(α-L-Arabinopyranosyl); RG: -(6-O-(6-deoxy-α-L-mannopyranosyl)-β-D- glucopyranosyloxy).

**Table 2 molecules-24-01661-t002:** Reversal effect of flavonoids on the cytotoxicity of daunorubicin to KB/MDR1 and KB cell lines.

No	KB/MDR1 Cells		KB Cells		RF
	IC_50_(μM)	RF_KB/MDR1_	IC_50_ (μM)	RF_KB_	
**C**	3.102 ± 0.441	1.000	0.715 ± 0.056	1.000	4.338
**E**	0.473 ± 0.005	6.818	0.455 ± 0.036	1.512	0.962
**1**	2.373 ± 0.970	1.307	0.993 ± 0.127	0.720	2.390
**2 ^t^**	1.579 ± 0.05	1.965	0.612 ± 0.107	1.169	2.582
**3**	1.580 ± 0.23	1.963	0.664 ± 0.051	1.077	2.380
**4 ^t^**	0.901 ± 0.042	3.443	0.955 ± 0.007	0.749	0.943
**5**	2.752 ± 0.211	1.127	0.905 ± 0.035	0.790	3.041
**6**	1.884 ± 0.243	1.646	2.610 ± 0.671	0.274	0.722
**7**	1.685 ± 0.623	1.841	0.916 ± 0.048	0.780	1.839
**8**	0.859 ± 0.137	3.613	0.413 ± 0.099	1.731	2.079
**9 ^t^**	0.676 ± 0.035	4.586	0.679 ± 0.023	1.053	0.996
**10**	3.226 ± 0.068	0.962	1.983 ± 0.078	0.361	1.627
**11**	5.894 ± 0.083	0.526	1.368 ± 0.077	0.523	4.310
**12**	5.501 ± 0.672	0.564	0.781 ± 0.133	0.916	7.046
**13**	1.459 ± 0.529	2.126	0.904 ± 0.158	0.791	1.613
**14**	1.560 ± 0.258	1.989	0.768 ± 0.113	0.931	2.030
**15**	2.891 ± 0.100	1.073	1.319 ± 0.041	0.542	2.192
**16**	2.463 ± 0.320	1.260	1.409 ± 0.225	0.508	1.748
**17**	5.353 ± 0.001	0.580	0.514 ± 0.006	1.391	10.414
**18**	1.214 ± 0.219	2.556	0.916 ± 0.064	0.781	1.325
**19**	2.976 ± 0.035	1.042	1.953 ± 0.197	0.366	1.524
**20 ^t^**	2.721 ± 0.067	1.140	0.567 ± 0.069	1.261	4.799
**21**	2.295 ± 0.054	1.352	1.312 ± 0.114	0.545	1.749
**22 ^t^**	2.803 ± 0.203	1.107	2.439 ± 0.189	0.293	1.149
**23**	2.076 ± 0.041	1.494	0.335 ± 0.015	2.138	6.205
**24**	2.084 ± 0.146	1.489	1.513 ± 0.057	0.473	1.377
**25**	2.371 ± 0.106	1.308	2.703 ± 0.182	0.264	0.877
**26**	2.611 ± 0.116	1.188	1.127 ± 0.092	0.635	2.317
**27 ^t^**	1.783 ± 0.055	1.740	0.962 ± 0.019	0.744	1.854
**28**	4.409 ± 0.540	0.704	0.203 ± 0.093	3.531	21.773
**29**	3.882 ± 0.172	0.799	2.393 ± 0.313	0.299	1.623
**30 ^t^**	1.267 ± 0.108	2.448	0.458 ± 0.072	1.561	2.767
**31**	2.986 ± 0.298	1.039	2.432 ± 0.227	0.294	1.228

C: control, negative control, without any inhibitors; E: elacridar, positive control; ^t^: Test set.

**Table 3 molecules-24-01661-t003:** The Pearson correlation between IC_50_ and related descriptors.

	IC_50_	vsurf_DW23	E_sol	dipole	vsurf_G
IC_50_	1.000	0.674 **	−0.432 *	−0.297	−0.041
vsurf_DW23		1.000	0.006	−0.268	−0.212
E_sol			1.000	−0.464 *	0.230
dipole				1.000	0.164
vsurf_G					1.000

*: Correlation is significant at the 0.05 level, **: Correlation is significant at the 0.01 level.

**Table 4 molecules-24-01661-t004:** Calculated results using the quantitative structure–activity relationship (QSAR) model.

No.	vsurf_DW23	E_sol	dipole	vsurf_G	IC_50_ (Experimental)	IC_50_ (Predicted)	Residuals
**1**	0.707	−3.559	0.841	1.252	2.373	2.180	0.193
**2**	1.000	−3.054	1.237	1.295	1.579	1.252	0.327
**3**	1.000	−3.711	0.896	1.288	1.580	2.490	−0.910
**4**	1.581	−2.211	1.348	1.326	0.901	1.026	−0.125
**5**	1.000	−1.574	0.972	1.351	2.752	2.149	0.603
**6**	3.391	−0.953	1.133	1.335	1.884	1.684	0.200
**7**	1.000	−4.867	1.158	1.244	1.685	1.612	0.073
**8**	0.707	−2.935	1.161	1.259	0.859	1.009	−0.150
**9**	1.000	−4.269	1.262	1.240	0.676	1.025	−0.349
**10**	1.000	−5.855	0.829	1.262	3.226	3.196	0.030
**11**	14.221	−5.762	0.769	1.275	5.894	5.914	−0.020
**12**	1.118	−11.881	0.997	1.322	5.501	5.494	0.007
**13**	0.707	1.216	0.776	1.409	1.459	2.336	−0.877
**14**	0.707	−0.596	0.679	1.256	1.560	1.670	−0.110
**15**	0.707	−0.369	0.420	1.293	2.891	2.808	0.083
**16**	0.500	1.410	0.440	1.275	2.463	1.874	0.589
**17**	11.597	−0.871	0.438	1.283	5.353	4.819	0.534
**18**	0.866	−3.328	1.024	1.284	1.214	1.878	−0.664
**19**	3.606	0.379	0.434	1.300	2.976	3.101	−0.125
**20**	0.500	−0.215	0.446	1.297	2.721	2.671	0.050
**21**	1.581	0.317	1.021	1.410	2.295	2.055	0.240
**22**	1.000	−2.655	0.523	1.258	2.803	2.982	−0.179
**23**	1.000	−2.064	0.551	1.264	2.076	2.740	−0.664
**24**	0.500	−0.711	0.690	1.293	2.084	2.032	0.052
**25**	1.118	−3.043	0.796	1.278	2.371	2.483	−0.112
**26**	1.000	−1.461	0.793	1.292	2.611	2.054	0.557
**27**	1.000	−3.440	1.426	1.416	1.783	2.071	−0.288
**28**	16.523	−1.351	0.657	1.244	4.409	4.782	−0.373
**29**	1.000	−4.664	0.811	1.339	3.882	3.652	0.230
**30**	0.500	−5.127	1.167	1.253	1.267	1.681	−0.414
**31**	1.000	−5.328	1.082	1.277	2.986	2.369	0.617

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
