# Peer review of "Quantitative Structure–Activity Relationships for the Flavonoid-Mediated Inhibition of P-Glycoprotein in KB/MDR1 Cells"

_molecules, 2019, doi:10.3390/molecules24091661_

Round 1
Reviewer 1 Report
Thank you for the revised form.
Reviewer 2 Report
The manuscript has been much improved and is in a nice condition now. I
recommend publication of this article.
Reviewer 3 Report
This manuscript has been properly revised.
This manuscript is a resubmission of an earlier submission. The following is a list of the peer review reports and author responses from that submission.
Round 1
Reviewer 1 Report
The authors performed a study to evaluate the flavonoid-mediated inhibition of the ABC transporter P-gp and described a 2D-QSAR model of P-gp inhibition based in their results, over 2 cell lines (KB and its parental KB/MDR1). Moreover, the P-gp inhibition was evaluated by IC50 of the P-gp substrate daunorubicin (MTT assay) in the presence or the absence of the flavonoids for 24h. The authors also compared the obtained results with P-gp inhibition by elacridar.
I have some concerns about this work:
- There are hundreds of studies reporting the P-gp inhibition by flavonoids. In fact, the authors cited some of them. Most of these studies described
ligand-based approaches (SAR, QSAR, 3D-QSAR and pharmacophore studies), and structure-based studies by using P-gp homology models. The authors need to explain clearly the innovation of this study.
- The use of daunorubicin as P-gp substrate has some challenges. In fact, it is also a substrate of BCRP and MRP (as admitted by the authors in line 83). Thus it seems that the results can be influenced by these ABC transporters. How can the authors assure the specificity of the P-gp inhibition?
Introduction section
- It was already described 4 generations of P-gp inhibitors, not only 3.
- There are different mechanisms of P-gp inhibition, the authors should describe them in the introduction section.
Methods
- Did the authors test the possibility of interference of each flavonoid in the MTT assay?
- The authors described the use of no cytotoxic concentration of the flavonoids. Did the authors also test the possibility of these compounds to promote cell proliferation?
Results
- A flavonoid showed an increase of P-gp function effect (line 81). Why the authors did not propose the possibility of induction or activation of P-gp by this flavonoids?
Discussion
- It seems to me that the discussion of the proposed 2D-QSAR model to each inhibition mechanism is too speculative based on the observed results
Reviewer 2 Report
This is a paper dealing with development of P-glycoprotein inhibitors to overcome multidrug resistance in cancer cells. Using an extensive 2D-QSAR method, the authors have identified flavonoid compounds to efficiently inhibit the activity of P-glycoproteins. The paper is well written, and the authors have clearly worked hard to produce a comprehensive dataset.
The authors should explain how the lack of 2, 3-double bond and 3’-OH, 4’-OH and increased number of methoxylated substitutions are beneficial to the inhibition of P-glycoprotein, illustrating the chemical structures of various flavonoids.
Reviewer 3 Report
Major comments: The authors built a 2DQSAR model and found that several descriptors were related to the inhibitory activity of flavonoids in the current experimental system. Several important points stated below need attention. (1) It would be desirable to indicate the CAS number of each compound in Table 1. Also, CAS numbers of elacridar and daunorubicin should be indicated. (2) I could not find Table 3 in the text. (3) How about the advantage of your 2DQSAR model comparing with other models? (4) The authors state that our model … activity of flavonoids (page 4, lines 105-106). This sounds somewhat redundant. (5) The authors state that vsurf_DW23 and … the ATPase activity (page 6, lines 170-171). This sounds somewhat of an overstatement.